# Challenges in identifying cancer genes by analysis of exome sequencing data

Matan Hofree[1,2,3,†], Hannah Carter[2,3,4], Jason F. Kreisberg[1,3], Sourav Bandyopadhyay[5], Paul S. Mischel[6], Stephen Friend[7] & Trey Ideker[1,2,3,4]

Massively parallel sequencing has permitted an unprecedented examination of the cancer exome, leading to predictions that all genes important to cancer will soon be identified by genetic analysis of tumours. To examine this potential, here we evaluate the ability of state-of-the-art sequence analysis methods to specifically recover known cancer genes. While some cancer genes are identified by analysis of recurrence, spatial clustering or predicted impact of somatic mutations, many remain undetected due to lack of power to discriminate driver mutations from the background mutational load (13–60% recall of cancer genes impacted by somatic single-nucleotide variants, depending on the method). Cancer genes not detected by mutation recurrence also tend to be missed by all types of exome analysis. Nonetheless, these genes are implicated by other experiments such as functional genetic screens and expression profiling. These challenges are only partially addressed by increasing sample size and will likely hold even as greater numbers of tumours are analysed.

[1] Cancer Cell Map Initiative (CCMI), 9500 Gilman Drive, La Jolla, California 92093, USA. [2] Department of Computer Science and Engineering, University of California San Diego, 9500 Gilman Drive, La Jolla, California 92093, USA. [3] Department of Medicine, University of California San Diego, 9500 Gilman Drive, La Jolla, California 92093, USA. [4] Moores Cancer Center, University of California San Diego, 3855 Health Sciences Drive, La Jolla, California 92093, USA. [5] Diller Family Comprehensive Cancer Center, University of California San Francisco, 1600 Divisadero Street, San Francisco, California 94115, USA. [6] Ludwig Institute for Cancer Research, University of California San Diego, 9500 Gilman Drive, La Jolla, California 92093, USA. [7] Sage Bionetworks, Seattle, 110 Fairview Avenue North, Seattle, Washington 98109, USA. † Present address: The Broad Institute of MIT and Harvard, 415 Main Street, Cambridge, Massachusetts 02142, USA. Correspondence and requests for materials should be addressed to T.I. (email: tideker@ucsd.edu).

Extensive collaborative projects[1–7] have released thousands of tumour exomes into the public domain, charting the complete sequences of the protein-coding regions[8]. Analysis of these data has revealed that each adult tumour carries ∼20–300 genes with somatic alterations to single nucleotides (single-nucleotide variants, SNVs)[9] or short (∼2–10) base insertions and deletions. In addition, tumours are perturbed by gene copy number variants (CNVs) and translocations of sequence, including aberrant fusion of two or more distinct genes within the same open reading frame (gene fusions)[10,11]. Given this landscape, an imminent challenge is to determine how best to interpret cancer genomic data, including which of the identified alterations promote the initiation or progression of cancer[10,12].

Towards this end, an increasing number of bioinformatic approaches are being developed with the goal of distinguishing true cancer 'driver genes' from genes randomly perturbed by 'passenger' mutations[9,13–17]. The main methodology has been to look for recurrent mutations within a cohort, based on how often a gene is altered relative to an expected background rate[9,13,14]. Other prominent methods score the predicted impact of a mutation on protein structure or function[15,16] or recognize spatial clustering of mutations within particular domains or residues[15,17]. Indeed, many of the best characterized cancer genes, such as *TP53*, *PTEN* and *PIK3CA*, are readily identified by any of these approaches[15]. Such success at recovering known cancer genes through exome analysis has led to recent estimates that we will soon obtain a comprehensive catalogue of all cancer genes if enough tumours are analysed, for example, 250,000 tumours covering 50 cancer types[18].

Accurate identification of all cancer genes is an ambitious task, however, and one can conceive of reasons why recovering all cancer genes might be difficult using the prevailing methods. First, despite many decades of study there remains no real consensus as to what exactly defines a 'cancer gene'. Most broadly, a cancer gene has been defined as any gene harbouring alterations that 'confer growth advantage on the cancer cell and are positively selected in the microenvironment of the tissue in which the cancer arises'[19]. Given this definition, it is unclear whether all such genes could be identified from analysis of somatic alterations in the exome, as alterations can also be selected from intergenic sequences[20], the germline[21] or, potentially, purely epigenetic causes[22]. To point, most known cancer genes (∼85%) were first identified through principles and experimental techniques other than somatic exome sequencing

(Fig. 1a,b). Examples include screens for human or viral DNA capable of transforming cells (for example, *RAS*)[23], systematic mutagenesis (for example, *PLEG1*)[24], differential expression or phosphorylation (for example, *TP53*)[25] or linkage of inherited germline variants to increased risk in familial cancers (for example, *BRCA1*)[26]. On the other hand, it is quite possible that cancer genes affected by one type of alteration also tend to be affected by other types, allowing them to be identified by multiple modes of analysis. For example, although TP53 was first identified based on differential expression, it was soon thereafter found to be recurrently mutated in the majority of cancers[25,27].

Second, the key task is not just to identify any genetic variant, but to classify variants that are functional from random passengers. Success at this classification task depends on the ability to cleanly discriminate cancer from non-cancer genes using some property, for example, mutation recurrence in a cohort. Previously[15,28], the focus has been to evaluate the proportion of genes identified in tumour sequence analysis that are likely to be cancer genes (precision) rather than the proportion of all cancer genes that are identified by a given method (recall). Achieving high recall may be difficult if some cancer genes are rarely altered, in which case an absence of statistically significant recurrence for a genetic alteration is not evidence that it lacks a functional effect.

Here we evaluate current tumour sequence analysis methods by their precision and recall for detecting known cancer genes. We find that these methods tend towards high precision but low recall, meaning that many true cancer genes are being missed by all methods. Estimates of statistical power suggest that this challenge is not likely to be addressed purely by increasing the number of tumour exomes analysed.

## Results

**Tumour exome analysis methods**. We considered five main methods, all of which have been explicitly designed with the goal of identifying cancer genes and were recently used in publications of The Cancer Genome Atlas (see http://www.nature.com/tcga/): MutSig Suite[9], OncodriveFM[29], OncodriveClust[30], ActiveDriver[17] and MuSIC[13]. Each of these approaches has been previously applied to analyse tumour exome data for 12 or more separate tissues and for a pooled 'pancancer' cohort, which combines all tissues into a single analysis (MAIN-METHODS, Table 1). The MutSig Suite tests each gene for enrichment of somatic SNVs versus a gene-specific background model. Enrichment is

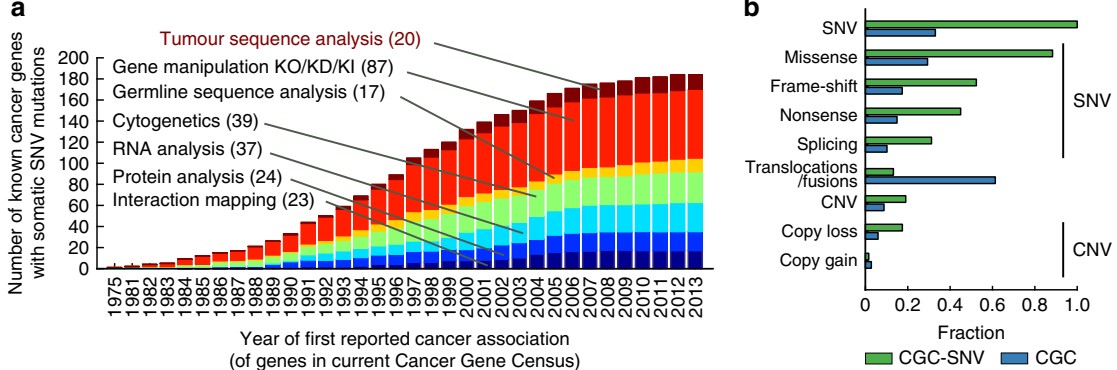

**Figure 1 | Original experimental techniques used to identify currently known cancer genes.** (**a**) Shown is the cumulative number of cancer genes known to be perturbed by somatic single-nucleotide variations, as recorded in the COSMIC CGC, according to the year of first cancer-related publication indexed in PubMed. Each bar is coloured by the experimental technique categories used by these first publications. In parenthesis is the number of genes associated with each experimental category as of 2013. (**b**) Proportion of the different types of somatic alteration included in the CGC. In blue are the proportions for all somatically altered genes; in green are the same proportions for genes also known to have single-nucleotide alterations.

**Table 1 | Prominent methods for cancer gene discovery by somatic exome analysis.**

| Method | Data type (method)* | Analysis principle | No. tissue cohorts (no. patients) | Total genes identified | Genes non-unique/unique to method | CGC non-unique/unique to method† | Ref. |
|---|---|---|---|---|---|---|---|
| *MAIN-METHODS* | | | | | | | |
| MutSig Suite | SNV (WES) | Combined (frequency, function, clustering) | 21 (4,742) | 260 | 191/69 | 98/7 | 9,18 |
| OncodriverFM | SNV (WES) | Function | 28 (6,792) | 426 | 281/145 | 127/31 | 29 |
| OncodriverCL | SNV (WES) | Clustering | 28 (6,792) | 79 | 72/7 | 52/2 | 30 |
| ActiveDriver | SNV (WES) | Clustering ( + phos-associated mutations) | 12 (3,205) | 106 | 74/32 | 30/5 | 15,17 |
| MuSIC | SNV (WES) | Combined (frequency, function, clustering, correlation with clinical phenotype) | 12 (3,205) | 182 | 141/41 | 81/3 | 13,15 |
| *ALT-METHODS* | | | | | | | |
| Gistic2.0—amplifications | CNV (SNP6) | Frequency | 34 (10,752) | 1,569 | 432/1137 | 53/21 | 14 |
| Gistic2.0—deletions | CNV (SNP6) | Frequency | 34 (10,752) | 6,897 | 671/6226 | 98/65 | 14 |
| IntOGen—CNV | CNV (SNP6) | Frequency + RNA expression | 16 (4,068) | 29 | 28/1 | 25/0 | 16 |
| Dendrix | SNV (WES) | Mutual exclusivity | 12 (3,281) | 17 | 28/2 | 23/1 | 32 |
| HotNet2 | SNV + CNV (WES + SNP6) | Network | 12 (3,281) | 147 | 96/51 | 43/0 | 31 |
| Fusion/translocations | FUS (RNA-seq) | Recurrent fusions | 13 (4,366) | 492 | 236/256 | 41/18 | 33 |
| | | TOTALS: | 42 | 8,871 | 906/7,967 | 175/153 | |

*Data types: CNV, copy number variant; FUS, gene fusion; SNV, single nucleotide variant. Methods: RNA-seq, RNA sequencing; SNP6, affymetrix SNP array; WES, whole-exome sequencing.
†Number of genes identified within the CGC-positive reference set.

measured in terms of three properties—SNV recurrence, impact on protein structure or function and spatial clustering—yielding a single $P$ value of significance integrating the three measures. OncodriveFM and OncodriveClust test for enrichment of SNVs for functional impact and spatial clustering, respectively. ActiveDriver examines clustering of SNVs in the active domains of kinase proteins. MuSIC scores enrichment of SNVs using the same three properties as the MutSig Suite but with a different background model.

We also considered five analysis methods that, although not specifically designed for cancer gene discovery, can be used for this purpose (ALT-METHODS, Table 1): Gistic2.0, a method for identifying focused genomic regions affected by recurrent CNVs (we used all genes within so-called 'narrow peaks')[14]; IntOGenCNV, which identifies genes within recurrent CNVs that have corresponding changes in messenger RNA (mRNA) expression level[16]; HotNet2, a method for mining public databases of molecular interactions to identify subnetworks of genes recurrently affected by SNVs and CNVs (we used all genes in 'core' and 'linker' networks)[31]; Dendrix, a method for detecting 'mutually exclusive' gene sets, in which SNVs are found in no more than one of these genes per patient[32]; and a method for detecting recurrent gene fusion events which we have labelled Fusion-genes[33]. All of the above methods pertain to somatic alterations to genes and, in particular, the MAIN-METHODS are based wholly on analysis of exome SNVs as the primary focus of our study. Some of the ALT-METHODS are assisted by additional measurements, for example, mRNA expression.

**Cancer gene reference lists**. To benchmark these methods, we considered multiple cancer gene lists compiled by various groups[34–37] (Table 2). The Cancer Gene Census version 73 (CGC) is a set of 571 genes manually curated by the Sanger Institute[34]. It includes genes affected by alterations of all types, including somatic and germline SNVs, CNVs and translocations; genes are annotated with the relevant cancer tissue types when this information is available. We considered the entire CGC as

well as the subsets of genes known to be impacted by the various alteration types (CGC-SNV, CGC-CNV, CGC-TRANS, CGC-SOMATIC and CGC-GERMLINE). Second, we queried UniprotKB, a manually curated database of protein functions[35], for the keyword-terms 'proto-oncogene,' 'oncogene' and 'tumour-suppressor gene,' resulting in 413 human genes. Third, a query of DISEASES[36], a database of disease-gene associations based largely on text-mining approaches, yielded a list of 691 genes associated with cancer. Fourth, the Atlas of Genetics and Cytogenetics in Oncology and Haematology (AGO) provides a set of 1,430 cancer genes manually curated by a collaborative effort spanning multiple centres[37]. We also assembled a list of negative control genes, that is, genes in which alterations are less likely to drive cancer. AGO provides such a negative set of 7,410 genes that have no evidence for association with cancer. We further filtered this list by excluding any gene in a cancer-related pathway from the MSigDB database[38], resulting in a conservative set of 2,217 negative control genes (AGO-NEG).

**Assessments of performance**. These benchmarks were used to evaluate each tumour sequence analysis method as follows. First, method performance was evaluated by considering the set of genes identified, whether by analysis of separate tissues or by pan-cancer analysis of all tissues together. We found that most methods were conservative in their detection, favouring precision over recall (Fig. 2a,b, Supplementary Fig. 1). Among the five MAIN-METHODS, testing against the CGC-SNV cancer reference and AGO-NEG negative control sets, recall ranged from 13% to a peak of 60% (Fig. 2c). In contrast, precision ranged from 59 to 90%. Performance results for other cancer genes sets were comparable or lower (Fig. 2a,b). Thus, the genes identified by tumour sequence analysis are likely to be known cancer genes (higher precision) but correspond to a smaller fraction of all known cancer genes (lower recall). As all methods had access to the same sets of reference cancer genes used for our benchmarks, and may have trained against these, these findings may represent an upper bound on performance.

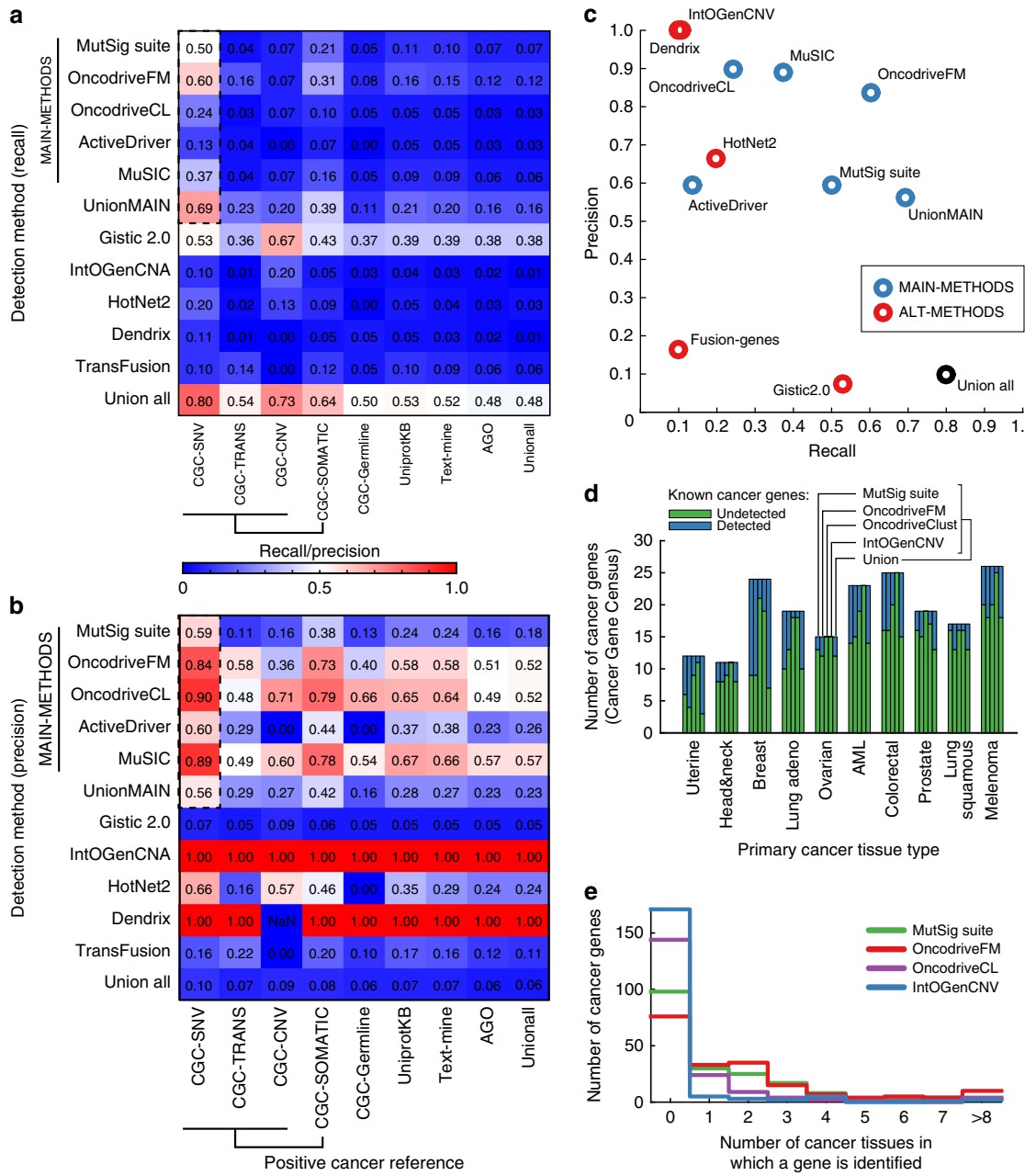

**Figure 2 | Performance of methods.** Heatmaps showing the (**a**) recall and (**b**) precision of each method (rows) tested against each positive cancer reference set (columns). Dashed box highlights the performance of MAIN-METHODS on the CGC-SNV reference set. To compute precision, we assume the proportion of cancer genes is 5% of all human genes; precision values for other proportions are shown in Supplementary Fig. 1 with qualitatively similar results. (**c**) Precision/recall plot detailing results from **a** and **b** for CGC-SNV cancer genes. (**d**) Summary of CGC-SNV genes curated for particular cancer tissues versus their cancer detection status based on genome analysis by four different methods and their union. (**e**) Count of CGC-SNV genes as a function of the number of cancer tissue types in which each gene has been detected thus far.

Next, we analysed the ability of the panel of methods to identify reference cancer genes specifically annotated to particular tissue types (Fig. 2d). For this purpose, we considered genes identified by a method when analysing data from each tumour tissue individually. Analysed in this way, recall was in the range of 0–75% with a median of 27%, depending on tissue and method (CGC-SNV set, Fig. 2d and Supplementary Fig. 2). Overall, we found that 59% of CGC-SNV reference genes annotated to a particular tissue were not identified by any of the methods applied to that tissue (Supplementary Fig. 2). On the other hand, of those identified, the majority were detected in more than one tissue type (Fig. 2e), which also held true when examining tumour

suppressor and oncogenes separately (Supplementary Fig. 3). Thus, analysis of tumour genome data from single tissues results in low recall characteristics that are qualitatively similar to those observed earlier in the aggregate analysis.

Of the 239 CGC cancer genes identified by any sequence analysis method, most were found by multiple methods (53% were by 2 or more, 27% by 4 or more). This observation led us to postulate that some cancer genes might be easier (or harder) to detect through all forms of sequence analysis. Further exploratory analysis revealed that genes identified by two or more methods tended towards high mutation frequency (one-sided Wilcoxon $P < 7.8 \times 10^{-13}$, Supplementary Fig. 4).

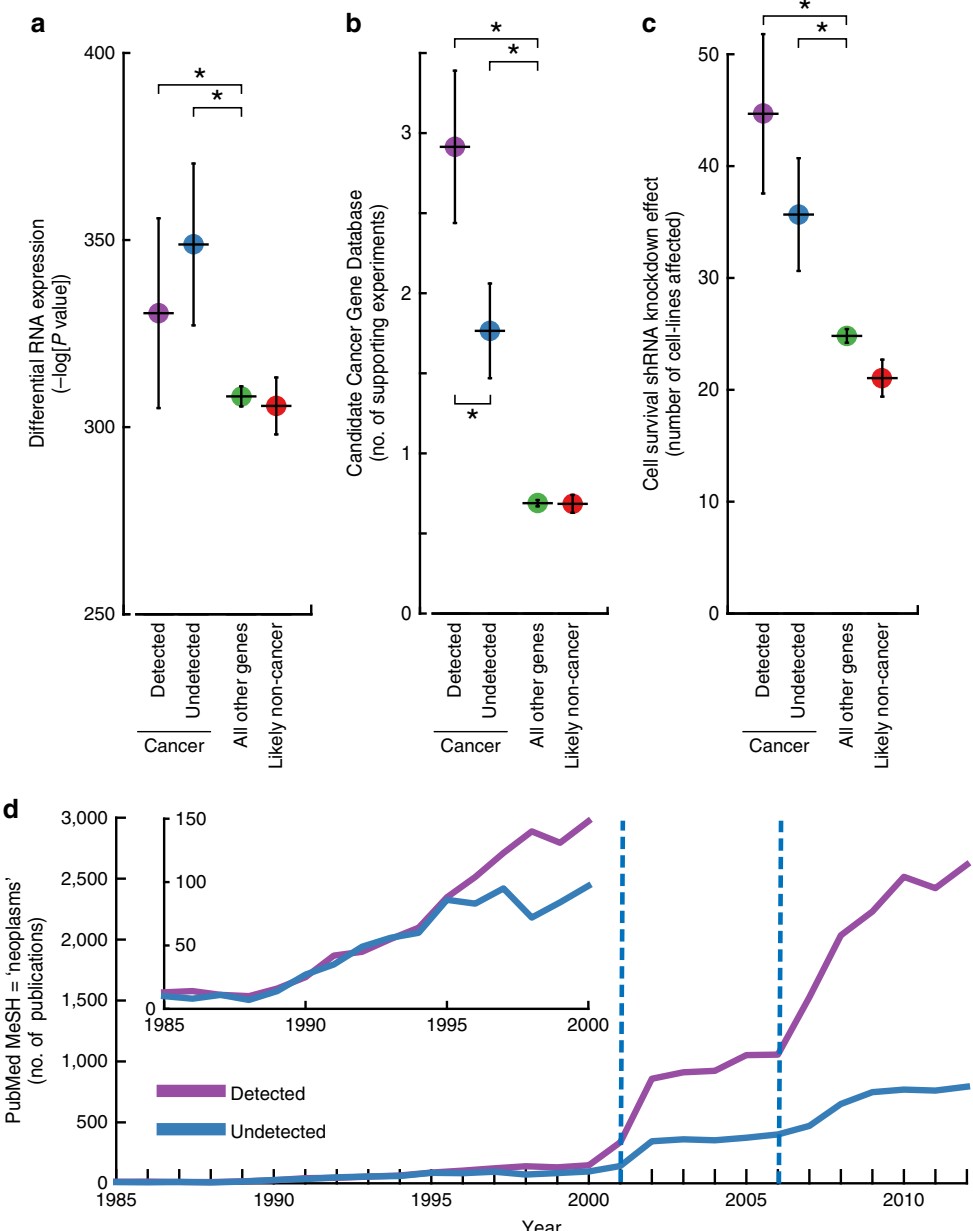

**Figure 3 | Experimental support for reference cancer gene lists.** (**a**–**c**) Support for CGC cancer genes detected by any of the MAIN-METHODS for analysing tumour genomes (Cancer Detected) versus those cancer genes that were undetected by any of these (cancer undetected). Also shown is support for the AGO-NEG negative control set of non-cancer genes (Likely non-cancer) and the remainder of genes in the genome-wide background (all other genes). Whisker plots indicate mean and the 95% confidence interval of the mean. Support is evaluated using: (**a**) RNA-seq tumour-normal differential expression in The Cancer Genome Atlas (TCGA). (**b**) Number of times a gene has been identified in independent cancer genetic screens in mice. (**c**) Number of Project Achilles cell lines with a measured impact (top/bottom 10%) on growth as a result of shRNA knockdown. An asterisk (*) indicates a significant difference in medians was found between the two sets. (**d**) The number of cancer publications by year comparing detected and undetected CGC cancer genes.

Moreover, a high mutation frequency was strongly associated with the performance of all methods, particularly in terms of recall, regardless of whether that method had been specifically designed as a mutation frequency detector (Supplementary Figs 5 and 6). Conversely, all methods tended to miss infrequently mutated genes. For example, methods such as MutSigCV identify genes based on increased frequency of SNVs compared with the genome-wide mutation rate, while methods such as oncodriveFM detect SNVs likely to be highly damaging to protein function; regardless, these methods show a very strong correspondence in the sets of cancer genes they identify[15,29]. Thus, genes that are commonly impacted by one type of event are likely to be impacted by other types of events also; conversely, genes rarely affected by an event type tend to be rarely affected by other types of events also. Given this consistent bias, it might be difficult for any method based solely on sequence analysis to recover some (for example, rarely mutated) cancer genes.

**Undetected reference genes have support in independent data.** A potential concern following these performance assessments was the validity of the cancer gene reference lists, as these lists might contain genes erroneously labelled as positive or negative cancer

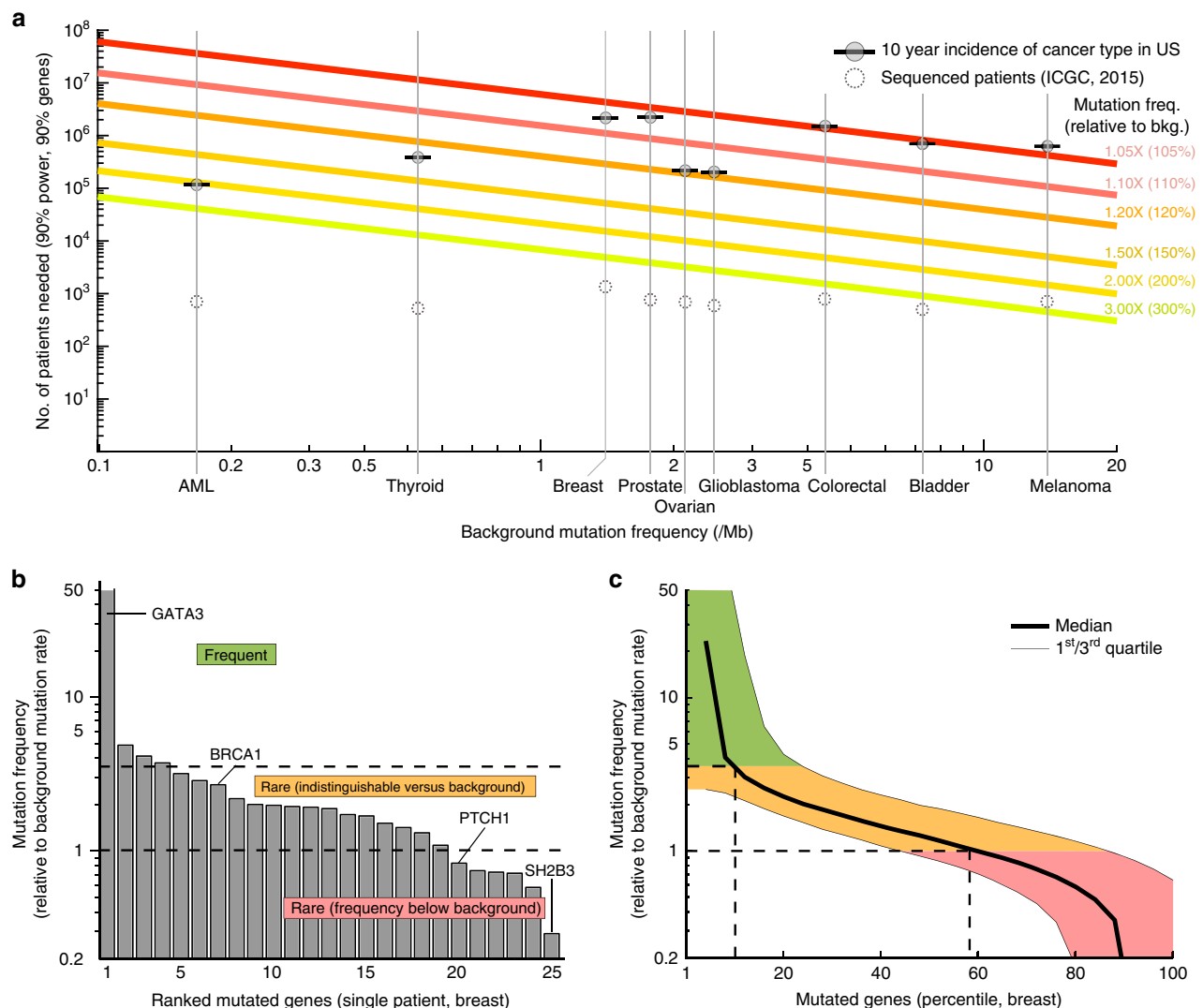

**Figure 4 | Power to detect recurrently mutated genes as the number of tumour exomes increases.** (**a**) Number of patient samples (*y* axis) necessary for detecting a cancer gene, as a function of the background somatic mutation rate of the tissue (*x* axis) and the fold increase in mutation rate of the cancer gene above this background (coloured lines). The total 10-year U.S. incidences of major cancer types are indicated (grey circles with horizontal bars), along with the number of patients currently sequenced as listed by the ICGC database v20 (dotted circles). (**b**) Mutated genes of a single breast adenocarcinoma patient, ranked by mutation frequency within tumours of this tissue type. (**c**) Same analysis showing the median behaviour for 881 The Cancer Genome Atlas (TCGA) patients with breast cancer. Mutated genes in each patient are ranked by mutation frequency; the median mutation frequency over all patients is plotted for each percentile.

genes. If a positive reference set were to include many genes with no functional role in cancer, or with low penetrance (that is, mutations that do not always lead to cancer), this situation could clearly explain the low recall of these methods. Accordingly, we examined the support of the reference sets according to experimental measures independent of somatic mutations, including tumour mRNA expression and functional genetic screening by gene knockout or knockdown. First, we observed that, as a whole, genes in the positive reference sets were significantly differentially expressed between tumour and normal samples. This was true of the genes detected by a MAIN-METHOD and also true of those undetected by any method (one-sided Wilcoxon $P < 5.6 \times 10^{-2}$ and $P < 4.0 \times 10^{-5}$, respectively, Fig. 3a and Supplementary Fig. 7). Second, we found that genes in the positive reference sets had orthologs that were significantly enriched in genetic screens for cancer drivers in mice, as previously determined by transposon-mediated gene knockout[39]. Once again, this result held true whether or not the

reference genes had been detected by sequence analysis methods (one-sided Wilcoxon $P < 1.49 \times 10^{-43}$ and $P < 3.06 \times 10^{-18}$, respectively; Fig. 3b and Supplementary Fig. 7). Third, both the detected and undetected cancer genes were strongly enriched for genes affecting cancer cell line growth, as previously determined by small hairpin RNA (shRNA) knockdown[40] (one-sided Wilcoxon $P < 4.5 \times 10^{-22}$ and $P < 2.15 \times 10^{-18}$, respectively, Fig. 3c and Supplementary Fig. 7). By all three of these measures, the negative control genes were found to be no different or significantly lower than background (two-sided Wilcoxon $P = 0.51$, $P = 0.23$ and $P < 0.02$ for differential expression, mouse genetic screens and cancer cell line genetic screens, respectively, for AGO-NEG). Although it is impossible to rule out any error in the reference sets, these findings suggest that they can be validated by multiple experimental screens performed independently of each other. Moreover, reference genes undetected by sequence analysis methods have a similar level of support as reference genes that are detected in terms of both

**Table 2 | Overview of positive cancer reference sets.**

|  | Number of genes | Curation process | Alteration type | Somatic/germline | Ref. |
|---|---|---|---|---|---|
| CGC-Somatic | 532 | Manual | SNV, CNV, Trans/fusion | Somatic | 34 |
| CGC-SNV | 188 | Manual | SNV | Somatic | 34 |
| CGC-TRANS* | 327 | Manual | Trans/fusion (not SNV) | Somatic | 34 |
| CGC-CNV* | 15 | Manual | CNV (not SNV) | Somatic | 34 |
| CGC-Germline† | 38 | Manual | SNV, CNV, Trans/fusion | Germline (not somatic) | 34 |
| UniprotKB | 412 | Manual | Unspecified | Both | 35 |
| Text-mining | 711 | Automated | Unspecified | Both | 36 |
| AGO | 1,430 | Manual | Unspecified | Both | 37 |

*Genes altered by translocations/fusions or CNVs, respectively, but not by SNVs.
†Genes altered in germline only; excludes genes also altered somatically.

differential expression and knockdown effect on cell line growth. Despite this support, undetected cancer genes have similar mutation frequencies as the negative reference genes or the genome-wide background (two-sided Kolmogorov–Smirnov test (KS-test) $P = 0.76$, Supplementary Fig. 8), suggesting a reason why they have not yet been found by any sequence analysis method.

We also found a large difference between cancer genes detected versus undetected by sequence analysis in terms of literature citations. Whereas both detected and undetected positive reference genes show a significantly elevated number of publications compared with background (one-sided Wilcoxon $P < 8.3 \times 10^{-44}$ and $P < 2.5 \times 10^{-16}$, respectively), far more publications are devoted to those that are detected by sequence analysis (one-sided Wilcoxon $P < 2.7 \times 10^{-22}$). In fact, genes in either category are cited with similar frequency in cancer publications through the late 1990's (Fig. 3d). Starting in 2001, however, the citation rate for detected cancer genes increases significantly, with a second rate increase starting in 2006 (F-test for both increases, $P < 2.3 \times 10^{-24}$). As these increases coincide with the emergence of human genomics and next-generation sequencing, respectively, one might speculate that DNA sequencing technology may have encouraged a focus in cancer research towards a particular class of 'sequence-detectable' cancer genes.

**Effects of increasing genome number and coverage.** Given the expected growth in DNA sequencing capacity over the next few years, it seems feasible that such information will soon be generated for a large proportion of cancer patients. Thus, a key remaining question was whether the low recall of exome sequence analysis methods might be addressed in the near future given much greater numbers of samples. Using a previously described power calculation[18], we estimated the minimum gene mutation frequency that can be distinguished accurately as a function of the number of tumour exomes analysed. By sequencing up to every new cancer patient in the United States, totalling $\sim 1.3$ million new cases annually[41], we found that the minimum detectable mutation frequency is indeed expected to decrease, from the present $\sim 3.4$ times the background mutation rate to $\sim 1.1$ (melanoma) to 2.0 (AML) times the background (Fig. 4a). Interestingly, nearly half of somatically mutated genes fall below that range (shown for breast cancer in Fig. 4b,c). These findings suggest that even if we analysed the exomes of all tumours in the US for 10 years using current methods, known cancer genes might remain undetected.

Another key question will be whether or not replacing exome sequencing with whole-genome sequencing (WGS) will improve the ability to identify certain cancer genes. For example, the promoter of the *TERT* telomerase gene has been found to be frequently mutated in cancer[42,43], allowing this gene to be clearly implicated by WGS. If many genes are like *TERT*, WGS might indeed pave the way for identifying a comprehensive catalogue of cancer genes. On the other hand, WGS greatly increases the number of background/passenger mutations under consideration and faces unique challenges such as associating mutations outside of coding regions with specific effects. While there are still too few cancer WGS studies to allow more than a cursory analysis, a recent WGS study of ovarian tumours reported no additional cancer genes beyond what had been found by earlier exome sequencing of the same cancer type[44].

An important distinction to be made is the difference between a recurrent alteration within a patient population (for example, *BRAF* V600E in melanoma), and a gene acting as a driver within a particular tumour under a specific context (for example, germline variants, tissue type and therapeutic regimen). While 'cohort drivers' are certainly interesting from a translational perspective, there is ample evidence for rare yet clinically important events[45–48]. Fundamentally, a complete compendium of cancer genes should include both cohort-recurrent alterations and individual, contextually-active drivers, including those that show a measurable effect only when analysed in combination.

From a clinical perspective, an individual patient cares not whether the cancer genes causing their tumour are frequently mutated in other patients, but whether their mutations can be targeted. Recurrent driver mutations are likely selected early during tumorigenesis[49], hence their critical role in many patients and potential as compelling drug targets in different cancer types. However, individual cancers may also be affected by rare mutations affecting the same pathways as common oncogenic drivers, and they too are selected in individual patients rendering them compelling personal drug targets. The findings presented here suggest that ignoring such mutations may miss important therapeutic opportunities. Indeed, cancer gene panels currently in use as clinical diagnostics[45,50–52] are based primarily on the most recurrently mutated genes in favour of more comprehensive alternatives. Before expanding the clinical panels to sequence all genes or all genomic DNA, we still need some method, or collection of methods, to distinguish which sequences are clinically important. Given the success of experimental techniques independent of sequence analysis (Fig. 1a), a viable strategy may be to complement ongoing genome sequencing campaigns with renewed efforts in cell biology, biochemistry, transcriptional profiling and other 'omics analysis to identify the key genes of cancer.

## Methods

**Tumour somatic mutation data.** We use somatic mutation data provided in the supplement of Lawrence *et al.*[4], downloaded from http://tumorportal.org on 22 January 2014, and available in Synapse (http://www.synapse.org), accession number syn1729383. These data contain tumour somatic variants and short insertion, and deletions for a set of 4,742 patients from 21 cancer types[18].

**Genes detected by each method.** The analysis presented here is exclusively based on results previously published for each of the respective methods, collected from publications listed in Table 1. Given the wide range of parameters and processing steps, we believe the best way to represent all methods is to use their own previously presented results from each publication.

**Analysis of literature support.** To evaluate literature support for each cancer gene (Fig. 1a), we use the NCBI gene2pubmed table (ftp.ncbi.nlm.nih.gov/gene/DATA/gene2pubmed.gz, 11 February 2015), from which we can compile a list of 1,015,815 publications (PubMed IDs) associated with human genes. The corresponding PubMed XML record for each publication is analysed to extract the associated MeSH subject terms, which are then indexed to the MeSH term tree (2015 version) propagating each association to all ancestor terms. We select publications that are MeSH-annotated as cancer related (Neoplasm;C04) with a corresponding experimental technique (Investigative Techniques;E05). The resulting mapping of PubMed publications to MeSH terms is used to compile all distinct cancer-related publications with an experimental component for each cancer gene. From this information we classify the first such publication of each gene into seven broad experimental categories by manual curation. Techniques or MeSH terms that cannot be assigned non-ambiguously are discarded (for example, PCR amplifications). To compare the amount of publications for cancer genes detected by any of MAIN-METHODS versus those left undetected, we examine all PubMed publications with cancer MeSH annotation (Neoplasm;C04) as above. To test for association between detected/undetected status and publication date, the number of publications per year (pubYear) is fit using a generalized linear model with three binary categorical variables: isDetected (detected or undected), isGenomeEra (published after the year 2000, the approximate era of genome sequencing) and isNextGenEra (published after 2006, the approximate era of next-generation sequencing). The optimal model is selected using a standard step-up procedure based on a Bayesian information criteria. This model: pubYear $\sim 1 +$ isDetected*isGenomeEra $+$ isDetected* isNextGenEra) includes a significant interaction between isDetected and isGenomeEra, and between isDetected and isNextGenEra (Z-test $P < 4.3 \times 10^{-3}$ and Z-test $P < 1.3 \times 10^{-5}$ respectively). A similar regression restricted to the period 1985–2000 fails to produce a model with any non-constant significant terms.

**Classification performance.** We use three measures of classification performance to compare the different methods for cancer gene detection. In each instance we use a positive and negative cancer reference and compare it with the cancer genes detected by a given method. The set of detected genes also found in the positive set is referred to as the true-positive (TP) set; similarly the set of detected genes not in the negative reference set is referred to as the false-positive (FP) set. The set of genes undetected in the negative set is referred to as the true-negative (TN) set; the set of undetected genes in the positive set is referred to as the false-negative (FN) set. Genes not included in either the positive or negative reference sets are ignored. To evaluate performance we use the following three measures: precision, $TP/(TP + FP)$; recall, $TP/(TP + FN)$; and informedness, recall $+$ specificity $- 1$.

**Differential expression analysis.** The Cancer Genome Atlas tumour and normal RNA-seq mRNA expression data are downloaded from Firehose (http://gdac.broadinstitute.org/, 14 February 2015) after normalization of RNA-seq reads to expected read counts using the RSEM algorithm and quantile normalization across all cancer tissue cohorts. We examine differential expression between tumour and normal samples in eight cohorts which each have $>40$ normal samples (breast, head/neck, clear cell renal, liver, lung adeno, lung squamous, prostate and thyroid), spanning a total of 540 normal and 4,816 tumour samples. We test for differential expression using a Welch's $t$-test for a difference in means of samples with unequal variances, and use the sum of the negative log $P$ values across the nine cohorts as a single summary statistic per gene.

**Response to shRNA knockdown.** Project Achilles shRNA response data (v2.4.3, 14 November 2014) provide RNAi knockdown values for 56,904 hairpin shRNA by 216 cell lines. These values are normalized using a simplified version of the ATARiS algorithm[53] resulting in knockdown values for 14,082 genes. For every gene we record the number of cell lines with knockdown values in the top or bottom 10%.

**Mutation frequency calculations.** Mutation frequency versus background is calculated using the MutSigCV method to detect enrichment in mutations[9]. Briefly, for every gene g and patient p, the non-synonymous (ns) mutation frequency $M_{p,g}^{ns}$ is the ratio of observed ns mutations $n^{ns}$ to expected ns mutations $E^{ns}$:

$$M_{p,g}^{ns} = \frac{n_{p,g}^{ns}}{E_{p,g}^{ns}} \qquad (1)$$

$E^{ns}$ is estimated separately for each gene using a local regression method as reported[18]. MutSigCV computes the $M_{p,g}^{ns}$ statistic for different classifications of mutations (transitions, transversions, nonsense, missense, etc.), then combines these different statistics to compute an overall $P$ value of significance that each gene

is frequently mutated in the patient cohort, assuming a beta-binomial null distribution. We base our analysis on the implementation and covariates provided as part of MutSigCV v1.4, downloaded on 22 June 2014. We have optimized this code for efficiency and to exploit parallel processing, but otherwise it is algorithmically identical to the original implementation.

**Estimating power to detect genes by mutation frequency.** Following a previously described procedure[18], we estimate the sample size needed to detect a frequently mutated gene at a genome-wide level of significance ($\alpha < 5 \times 10^{-6}$) with power at least 90% ($\beta \geq 0.9$). Effect size ($\varphi$) is computed as the difference in proportions between the mutation rate of the cancer gene ($P_1$) and the background mutation rate of similar genes in that tissue type ($P_0$). Given $\alpha$, $\beta$ and $\varphi$, sample size is computed using a standard method[54]. $P_0$ is defined as described[18]:

$$P_0 = 1 - \left(1 - \mu f_g\right)^{\frac{3L}{4}} \qquad (2)$$

$\mu$ A variable representing the background mutation frequency of the tissue.

$f_g = 3.9$ A gene-specific background mutation rate estimated by MutSigCV. The selected value corresponds to the 90 percentile of genes.

$L = 1,500$ The length of a gene in coding bases, representing the 90 percentile of genes.

$\frac{3}{4}$ Corresponds to the typical fraction of mutations that are non-synonymous.

$R$ A variable representing the mutation frequency above background.

$P_1$ frequency is defined as a multiplicative product of the background:

$$P_1 = (1 + r)P_0 \qquad (3)$$

**Data availability.** Data used in this study was gathered from publicly available sources as indicated above. Literature support and year of first mention in cancer publication for each cancer gene was downloaded from NCBI gene2pubmed table (ftp://ftp.ncbi.nlm.nih.gov/gene/DATA /gene2pubmed.gz, downloaded on 11 February 2015). Cancer genes detected by each method were downloaded from the Supplementary Information and accompanying data provided along with publications referenced in Table 1. Known cancer reference genes were downloaded from the sources indicated in Table 2, in particular, Sanger Institute Cancer Gene Censuses database (CGC, http://cancer.sanger.ac.uk/files/cosmic/current_release/cancer_gene_census.csv, version 73, downloaded on 28 June 2015). Tumour expression data were downloaded from the Firehose repository (http://gdac.broadinstitute.org/, downloaded on 14 February 2015). Response to shRNA knockdown measurements were downloaded from the Project Achilles data portal (http://www.broadinstitute.org/achilles/datasets/5, version 2.4.3, download on 14 November 2014). Mouse genetic screening data were downloaded from the CCGD database (http://ccgd-starrlab.oit.umn.edu/, downloaded on 19 August 2015). Tumour somatic mutation data for individual patients is available on Synapse (http://www.synapse.org), accession number syn1729383. All other data is contained within the Article and Supplementary Information, or available from the authors on request.

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

## Acknowledgements

This work was supported by the National Institutes of Health (R01 ES014811 and U24 CA184427) and a generous donation from the Fred Luddy Family Foundation. We thank Billur H. Engin for contributions that were not included in this work. We further wish to thank Anne-Ruxandra Carvunis, John Paul Shen, Andrew Gross, James Jensen, Michael Kramer and Eran Hodis for insightful comments and suggestions on the manuscript and figures.

## Author contributions

M.H. and T.I. designed the study. M.H. performed the statistical analyses. M.H., H.C. and T.I. analysed the data. M.H., J.F.K. and T.I. wrote the manuscript, with contributions from H.C., S.B., P.S.M. and S.F.

## Additional information

**Competing financial interests:** The authors declare no competing financial interests.

**How to cite this article**: Hofree, M. *et al.* Challenges in identifying cancer genes by analysis of exome sequencing data. *Nat. Commun.* **7**:12096 doi: 10.1038/ncomms12096 (2016).

