## [Peer Review File · Nature Communications]

Reviewer #1 (Remarks to the Author):

Hofree et al. present a revised version of their manuscript entitled 'Challenges in identifying cancer genes by tumor genome analysis'. The authors have addressed some, but not all of my criticism.

The two main issues with the manuscript were and to some extent remain the following:

1. There is only a vague definition of the term 'cancer gene', in particular with a view on what type of genomic change causes pathogenicity in a given gene. In that light I think it is mandatory to clearly distinguish genes affected by point mutations, copy number changes, rearrangements and also other cancer genes with a different proposed oncogenic mechanism, such as overexpression.

I don't think that being blunt about this helps to advance the field, in particular because different types of genomic lesions require different assays for their detection (see below and my previous comments). Therefore, it appears imperative to show the proposed mechanisms of pathogenicity for CGC and also the other lists and discuss this upfront.

I think that this is also a missed opportunity to critically assess the colloquial definition of a cancer gene, because the alternative explanation of the authors' main findings - a low recall among all cancer gene lists and methods used - is that many authors simply use a too broad definition of the term 'cancer gene'.

Note: The authors present a figure showing the type of CGC lesions in the rebuttal, but figure 1 in the manuscript is still the historical approach for discovery.

2. In combination with the above vagueness on the definition of a cancer gene, the authors should be more explicit what type of data their analysis is based on and still rely for the most part on the rather broad term 'genome analysis' in the title, abstract and most of the paper. It is unclear what this term refers to and it suggests that the authors had assessed whole-genome sequencing data, which is not the case.

It should instead be clearly stated in the abstract that the authors use whole exome sequencing, SNP6 arrays and gene expression data for their assessment, which has obvious limitations.

Given these more restricted genomic assays it is, however, no surprise that the analysis methods used here struggle to detect gene fusions, which make up half of the Cancer Gene Census, and by the choice of tools also do not recover genes with differential expression, which is one of the more commonly described mechanisms in the Atlas of Genetics and Cytogenetics in Oncology and Haematology.

The benefits of WGS over exome sequencing include better power for detecting medium sized indels, rearrangements and focal copy number changes. This should also be discussed. Most likely WGS will also not recover all genes, but I would expect it to yield a substantially better recall, also due to the ability of better quantifying background mutation rates - see reviewer 4's comment.

In summary, the combination of data types, analysis tools and content of the cancer gene lists implies that one can only realistically expect a decent recall for the CGC-SNV comparison, which the focus should be on. Given the two limitations outlined above I don't think that the claim that 'tumour genome' analysis misses half of the known cancer genes is well supported.

Minor:

1. A query to UniprotKB with keyword "oncogene" "proto-oncogene" "tumor suppressor" yields 1,749 genes

[http://www.uniprot.org/uniprot/?query="tumor+suppressor"+OR+oncogene+OR+"proto+oncogene"&fil=reviewed%3ayes+AND+organism%3a"Homo+sapiens+\(Human\)+%5b9606%5d"&offset=300&sort=score&columns=id%2centry+name%2creviewed%2cprotein+names%2cgenes%2corganism%2clength](http://www.uniprot.org/uniprot/?query=)
Please clarify.

2. Showing differential expression / similar lethality for missed CGC genes confirms that these are more likely to be cancer genes, but these are missed as they are mostly gene fusions and absent in exome data. How do these comparisons look like for the genes missed in other 'cancer gene' lists, which could include genes affected by epigenetic or other non-genomic mechanisms.

Reviewer #4 (Remarks to the Author):

The authors comprehensively addressed the questions raised by the reviewers. The limitations of the study (e.g. focus on WES)) are now addressed in the main manuscript. Analysis with additional gold standards has been added and a the correlation between cohort sizes and recall has been performed, strengthening the claim of the authors.

While I am still not convinced that the limitations of Exome-seq cannot be overcome by a combination of WGS, RNA-seq and Epigenome sequencing methods in large cohorts, the authors correctly state that the performance of these sequencing techniques remains to be evaluated and is currently only measurable in a few cohorts. Therefore the manuscript facilitates an important discussion about the limitations of current sequencing methods and supports the move from simple Exome analysis to integrative omics and network analysis methods in cancer research.

Hofree et al. “Challenges in identifying cancer genes by tumor genome analysis” Point by point response to reviewer comments

Reviewer #1:

Hofree et al. present a revised version of their manuscript entitled 'Challenges in identifying cancer genes by tumor genome analysis'. The authors have addressed some, but not all of my criticism. The two main issues with the manuscript were and to some extent remain the following:

1. There is only a vague definition of the term 'cancer gene', in particular with a view on what type of genomic change causes pathogenicity in a given gene.

We have rewritten the manuscript to provide additional clarity both in the definition of cancer genes and (per this reviewer’s second comment below) in the types of analysis methods being considered. The main additions are highlighted in the revised manuscript, starting with the text in paragraph 3:

“First, despite many decades of study there remains considerable controversy as to what exactly constitutes a cancer gene. Most broadly, a cancer gene has been defined as any gene harboring alterations that “confer a selective growth advantage to cells within a particular microenvironment” (Stratton 2009).

A fundamental goal of our paper is to address the question: ‘To what extent can cancer genes (of various types A) be identified by genetic analysis (of types B)?’, where:

A = Types of molecular alterations providing a selective growth advantage

B = State-of-the-art methods, all based on different ways of analyzing the somatic exome

Importantly, recent papers have suggested that particular genetic analysis methods in B are able to find all cancer genes, with the exact definition of A underspecified. Our manuscript evaluates this prospect in the context of a broader survey of the existing options for A and B. We thank the reviewer for helping us achieve more clarity in this respect.

In that light I think it is mandatory to clearly distinguish genes affected by point mutations, copy number changes, rearrangements and also other cancer genes with a different proposed oncogenic mechanism, such as overexpression. I don't think that being blunt about this helps to advance the field, in particular because different types of genomic lesions require different assays for their detection (see below and my previous comments). Therefore, it appears imperative to show the proposed mechanisms of pathogenicity for CGC and also the other lists and discuss this upfront.

We have added a new table (Table 2) and a new figure panel (Fig. 1B) that describe the breakdown of the gold-standard cancer gene lists (CGC and others) in terms of known oncogenic mechanisms such as point mutation, copy number changes, and rearrangements and the respective performance of the different methods on these. Also, we now show explicitly the precision/recall of each method on each subset of the CGC list (CGC-SNV, CGC-CNV, CGC-TRANS, CGC-Somatic, CGC-Germline in new Figure 2A,B). Also as requested, these various alteration types are now discussed explicitly up front, in the Introduction and again in paragraph six where the gold-standard cancer gene lists are described (see highlighted text in revised manuscript).

I think that this is also a missed opportunity to critically assess the colloquial definition of a cancer gene, because the alternative explanation of the authors' main findings - a low recall among all cancer gene lists and methods used - is that many authors simply use a too broad definition of the term 'cancer gene'.

We agree that the field would benefit from an explicit discussion of what defines a cancer gene. See our comments above. We hope that the revised manuscript will facilitate this discussion.

Note: The authors present a figure showing the type of CGC lesions in the rebuttal, but figure 1 in the manuscript is still the historical approach for discovery.

We have now added a new Fig. 1B showing the type of CGC lesions, as requested. We have also exchanged the original Figure 1 (now Fig. 1A) to focus only on cancer genes that have been previously associated with point mutations (CGC-SNV). This shift does not fundamentally change the finding that most of these known cancer genes were originally implicated by methods other than tumor sequence analysis.

2. In combination with the above vagueness on the definition of a cancer gene, the authors should be more explicit what type of data their analysis is based on and still rely for the most part on the rather broad term 'genome analysis' in the title, abstract and most of the paper. It is unclear what this term refers to and it suggests that the authors had assessed whole-genome sequencing data, which is not the case. It should instead be clearly stated in the abstract that the authors use whole exome sequencing, SNP6 arrays and gene expression data for their assessment, which has obvious limitations.

We agree and have updated the Abstract to clearly state that we survey the current state-of-the-art methods, which are based on analysis of the somatic exome. We have organized the methods into MAIN-METHODS and ALT-METHODS: all MAIN-METHODS are based on analysis of SNVs; some ALT-METHODS draw information from other data layers, such as use of mRNA transcripts to detect gene fusions or assist in the identification of exons impacted by CNVs. The sources of data for each method are clarified in the text and in Table 1. All methods, including the ALT-METHODS, pertain to somatic alterations of genes. We feel this is a reasonable scope and organization for our

survey, especially given the need to evaluate statements made by others about the ability of a subset of these methods to “identify all cancer genes.” We mention in the Discussion that future analysis methods based on other data types, e.g. whole-genome sequences, may be able to improve our ability to identify cancer genes. As noted by Reviewer 4, we do feel that our survey stimulates “an important discussion about the limitations of current sequencing methods.”

Given these more restricted genomic assays it is, however, no surprise that the analysis methods used here struggle to detect gene fusions, which make up half of the Cancer Gene Census, and by the choice of tools also do not recover genes with differential expression, which is one of the more commonly described mechanisms in the Atlas of Genetics and Cytogenetics in Oncology and Haematology.

We could not agree more -- methods based on somatic SNV analysis may miss cancer genes if they are altered mainly by gene fusions, gene expression, etc. However, based on recent publications and associated claims that certain methods find “all cancer genes,” we feel that a serious discussion needs to take place. Part of the confusion is almost certainly due to reasons this reviewer raises: lack of clarity in the definition of cancer gene, including the different types of lesions considered, or in the types of data used for detection. We thank the reviewer for helping us hone these points in our manuscript.

There is also an important question of whether cancer genes originally identified by one type of lesion can also be recognized by other types. In particular, we now examine systematically whether, generally, a cancer gene of type A can be identified by an analysis of type B (Figure 2A,B and Supplemental Figure 1).

In terms of gene fusions, a prominent published method for detecting these is already included in our survey (ALT-METHODS).

The benefits of WGS over exome sequencing include better power for detecting medium sized indels, rearrangements and focal copy number changes. This should also be discussed. Most likely WGS will also not recover all genes, but I would expect it to yield a substantially better recall, also due to the ability of better quantifying background mutation rates - see reviewer 4's comment.

As requested, we have added text to our Discussion (**highlighted in the manuscript**) to point out the potential benefits of WGS over exome sequencing.

In summary, the combination of data types, analysis tools and content of the cancer gene lists implies that one can only realistically expect a decent recall for the CGC-SNV comparison, which the focus should be on. Given the two limitations outlined above I don't think that the claim that 'tumour genome' analysis misses half of the known cancer genes is well supported.

In fact, convinced by some of the reviewer's points, we have now softened the claim in the Abstract that tumor genome analysis misses half of known cancer genes. We instead make this point more precisely, as follows:

“While many cancer genes are identified by analysis of the recurrence, spatial clustering or predicted impact of somatic mutations, we find that many such genes remain undetected due to lack of discriminative power (13-60% recall of cancer genes impacted by somatic single nucleotide variants, depending on method).”

Minor:

1. A query to UniprotKB with keyword "oncogene" "proto-oncogene" "tumor suppressor" yields 1,749 genes

[http://www.uniprot.org/uniprot/?query="tumor+suppressor"+OR+oncogene+OR+"proto+oncogene"&fil=reviewed%3Ayes+AND+organism%3A"Homo+sapiens+\(Human\)+%5B9606%5D"&offset=300&sort=score&columns=id%2Centry+name%2Creviewed%2Cprotein+names%2Cgenes%2Corganism%2Clength](http://www.uniprot.org/uniprot/?query=)

The query below (performed on March 4th 2016) results in a set of 413 genes:

[http://www.uniprot.org/uniprot/?query=keyword:%22Tumor%20suppressor%20\[KW-0043\]%22%20OR%20keyword:%22Proto-oncogene%20\[KW-0656\]%22%20OR%20keyword:%22Oncogene%20\[KW-0553\]%22&fil=reviewed%3Ayes+AND+organism%3A%22Homo+sapiens+%28Human%29+%5B9606%5D%22&sort=score](http://www.uniprot.org/uniprot/?query=keyword:%22Tumor%20suppressor%20[KW-0043]%22%20OR%20keyword:%22Proto-oncogene%20[KW-0656]%22%20OR%20keyword:%22Oncogene%20[KW-0553]%22&fil=reviewed%3Ayes+AND+organism%3A%22Homo+sapiens+%28Human%29+%5B9606%5D%22&sort=score)

To clarify this matter we now include all gold-standard reference sets in an additional Table 2.

2. Showing differential expression / similar lethality for missed CGC genes confirms that these are more likely to be cancer genes, but these are missed as they are mostly gene fusions and absent in exome data. How do these comparisons look like for the genes missed in other 'cancer gene' lists, which could include genes affected by epigenetic or other non-genomic mechanisms.

We now include an additional supplementary figure showing similar differences in experimental evidence for the CGC-SNV, UniprotKB, and AGO gold-standard reference sets (Supplementary Figure 7).

Reviewer #4:

The authors comprehensively addressed the questions raised by the reviewers. The limitations of the study (e.g. focus on WES)) are now addressed in the main manuscript. Analysis with additional gold standards has been added and a the correlation between cohort sizes and recall

has been performed, strengthening the claim of the authors. While I am still not convinced that the limitations of Exome-seq cannot be overcome by a combination of WGS, RNA-seq and Epigenome sequencing methods in large cohorts, the authors correctly state that the performance of these sequencing techniques remains to be evaluated and is currently only measurable in a few cohorts. Therefore the manuscript facilitates an important discussion about the limitations of current sequencing methods and supports the move from simple Exome analysis to integrative omics and network analysis methods in cancer research.

We thank the reviewer for this positive assessment of our revised manuscript.

Reviewer #1 (Remarks to the Author):

Hofree et al. present a revised version of their manuscript "Challenges in identifying cancer genes by tumor genome analysis".

In the current manuscript the authors compare state of the art methods for detecting cancer genes based on publicly available whole exome sequencing, SNP6 copy number variation and RNA-seq data from more than 3,000 cancer patients from TCGA. The authors evaluate the precision and recall of each method to rediscover different classes of cancer genes from the COSMIC Cancer Gene Census and other sources. For the subset of CGC genes altered by point mutations (approximately 50% of CGC) the authors observe a recall of below 60% for individual methods, which might be increased to 70% taking the union, albeit at lower precision (about 60%). A similar recall is reported for genes affected by copy number alterations, but most gene fusion events are missed in the absence of whole genome sequencing data. The authors suggest that there may be signal in other data types (gene expression/mouse screens/RNAi) that could help the discovery of missing cancer genes. Lastly, the authors contrast the number of samples needed for detecting cancer genes in different cancer types with the corresponding 10-year cumulative incidence, indicating that genes with low recurrence may be missed by current methods even if all cases in a decade were sequenced.

Compared to previous submissions, the authors have addressed my main concern and are now more explicit about the mechanism by which cancer genes are impaired (point mutations/copy number alterations/gene fusions, Table 2, Figure 1B) and report separate values for each cancer gene subset and method in Figure 2A-B. This adds much clarity to the interpretation of their findings and highlights the strength and weaknesses of individual methods for particular cancer gene class.

I have no other concerns.

Moritz Gerstung European Bioinformatics Institute EMBL-EBI